# Adsorptive Removal of Azithromycin Antibiotic from Aqueous Solution by Azolla Filiculoides-Based Activated Porous Carbon

**DOI:** 10.3390/nano11123281

**Published:** 2021-12-03

**Authors:** Davoud Balarak, Amir Hossein Mahvi, Saeideh Shahbaksh, Md A. Wahab, Ahmed Abdala

**Affiliations:** 1Department of Environmental Health, Health Promotion Research Center, Zahedan University of Medical Sciences, Zahedan 9816743463, Iran; 2Center for Solid Waste Research, Institute for Environmental Research, Tehran University of Medical Sciences, Tehran 1417653911, Iran; ahmahvii@gmail.com; 3Student Research Committee, Zahedan University of Medical Sciences, Zahedan 9816743463, Iran; dbchemistry2@gmail.com; 4Institute for Advanced Study, Chengdu University, Chengdu 610106, China; 5Chemical Engineering Program, Texas A&M University at Qatar, Doha P.O. Box 23874, Qatar

**Keywords:** Azolla filiculoides, activated porous carbon, azithromycin, adsorption equilibrium, thermodynamics, adsorption, water treatment

## Abstract

Due to the shortage of freshwater availability, reclaimed water has become an important source of irrigation water. Nevertheless, emergent contaminants such as antibiotics in reclaimed water can cause potential health risks because antibiotics are nonbiodegradable. In this paper, we report the adsorptive removal of azithromycin (AZM) antibiotics using activated porous carbon prepared from Azolla filiculoides (AF) (AFAC). The influence of the adsorption process variables, such as temperature, pH, time, and adsorbent dosage, is investigated and described. The prepared AFAC is very effective in removing AZM with 87% and 98% removal after the treatment of 75 min, at 303 and 333 K, respectively. The Langmuir, Temkin, Freundlich, and Dubinin–Radushkevich isotherm models were used to analyze the adsorption results. The Freundlich isotherm was best to describe the adsorption isotherm. The adsorption process follows second-order pseudo kinetics. The adsorption was endothermic (Δ*H*°= 32.25 kJ/mol) and spontaneous (Δ*S*° = 0.128 kJ/mol·K). Increasing the temperature from 273 to 333 K makes the process more spontaneous (Δ*G*° = −2.38 and −8.72 KJ/mol). The lower mean square energy of 0.07 to 0.845 kJ/mol confirms the process’ physical nature. The results indicate that AFAC can be a potential low-cost adsorbent of AZM from aqueous solutions.

## 1. Introduction 

Pharmaceuticals play a critical role in saving millions of lives from various infectious to deadly diseases and lengthening life spans. On the other hand, pharmaceutical contaminants in water sources (surfaces, lakes, rivers, and seas), effluents of wastewater treatment plants, soils, and sludges, are emerging contaminants, leading to chronic and acute effects on the environment and human life [1,2]. Wastewater effluents are widely discharged from different sectors, such as industries, households, hospitals, and pharmaceuticals factories [3,4]. These streams may contain traces of various pharmaceuticals that might lead to bioaccumulation in the environment and endanger humans and animals [5]. Pharmaceuticals are a diverse group of chemical compounds with varying physical and chemical properties [6]. Therefore, various pharmaceuticals’ successful use has the drawback of their emergence as rapidly growing harmful contaminants.

Among the pharmaceutical products, antibiotics are a critically important class of pharmaceuticals extensively used to prevent and cure infectious and deadly diseases in human and veterinary applications. Among the antibiotics, azithromycin (AZM) is a relatively new macrolide employed in eliminating erythromycin shortcomings, such as intolerance and its limited antimicrobial spectrum. AZM is more widely used as an effective antibiotic for several bacterial infections, including strep throat, middle ear infections, traveler’s diarrhea, pneumonia, and intestinal disorders [7]. AZM is also employed to treat several sexually transmitted diseases, including chlamydia and gonorrhea infections [8]. In addition, AZM could function along with other medications to treat malaria, with doses administered orally or intravenously [9]. Antibiotics are usually not removed by the primary and secondary water treatment steps [10]. Therefore, they can exist in reclaimed water, developing antibiotic-resistant strains and threatening human health and the ecosystem [11,12]. A large portion of antibiotics is excreted as intact from the body, as antibiotics are usually not biodegradable and can survive in aquatic environments for a long duration. As a result, antibiotics negatively impact the environment, such as creating toxicity for infection pathogens, including bacteria, even at very low concentrations, disturbing biological processes in wastewater treatment, and decreasing the degradation rate of organic compounds. The discharge of these streams raises bacterial resistance to antibiotics and negatively impacts the environment. Therefore, the removal of them has become a crucial issue.

Meanwhile, different processes on removing antibiotics from the aquatic environment, including reverse osmosis, adsorption, biodegradation, chemical oxidation, solvent extraction, and nanoemulsion [13,14,15,16], have been demonstrated. Recently, the green nanoemulsion method was used to treat polluted water with AZM [17,18]. Among the processes, adsorption as a wastewater treatment process has become an attractive method because it offers the advantages of easy preparation and handling, low cost, high efficacy, and low or no risk of producing toxic byproducts [19]. 

Notably, the adsorption process is primarily affected by the adsorbent type and surface-structural properties, including surface area, pore size, porosity, and pore volume. Carbonaceous material-based adsorbents are well-recognized for solving such environmental issues. Among these carbonaceous adsorbents, activated carbon (AC) was one of the effective adsorbent materials. In this context, an inexpensive AC adsorbent is demanded to reduce the adsorption process cost [18,19]. The cost of AC production depends mainly on its precursor [20]. AC production was investigated using inexpensive materials, such as waste rice hulls, waste potato residue, seaweed, and almond shells [21,22]. 

Azolla filiculoides (AF) have recently been explored as a candidate for activated porous carbon production due to their relatively high carbon content and meager cost [23,24,25]. Therefore, AF has received tremendous attention when it comes to removing different pollutants [26,27]. AF can grow on wastewater and adsorb inorganic and organic materials from waste effluents [28]. With the recent interest in the rapid growth of AF [29,30], this alga, as an inexpensive and readily available adsorbent, should significantly alleviate the severity of this problem [27]. However, the percentage of removal using alga as an adsorbent is low. Therefore, activated carbon prepared from alga can increase the removal percentage to a significant amount [31], which has hardly been explored. Therefore, this study reports on the facile preparation of porous AC from AF (AFAC) and its performance for the removal of AZM. AFAC was analyzed by scanning electron microscope (SEM), N_2_ adsorption measurements, and X-ray photoelectron spectroscopy (XPS). The main characteristics of AZM adsorption, including adsorption isotherms, adsorption kinetics, and adsorption thermodynamics, are also analyzed.

## 2. Materials and Methods

### 2.1. Materials 

AZM dihydrate (C_38_H_72_N_2_O_12_·2H_2_O, the structure is provided in Figure 1) (Analytical grade) from Sigma Aldrich (Darmstadt, Germany), and research-grade acetonitrile from Merck (Darmstadt, Germany), were purchased and used without further purification. The AC precursor, AF, was obtained from the Anzali wetland (Anzali City, Iran).

### 2.2. Preparation of Activated Porous Carbon from AF

The washed and dried AF was milled into a fine powder, sieved through 0.45 mm stainless steel mesh, and immersed in 28% ZnCl_2_ aqueous solution for 12 h, followed by drying at 105 °C overnight. The zinc-treated AF was then pyrolyzed under N_2_ atmosphere in a tube furnace at 350 °C for 2 h and then at 600 °C for 4 h. The produced porous AC (AFAC) was washed with hot 0.5 M HCl, filtered, and repeatedly rinsed with warm water until no free zinc ions were detected. The final AFAC product was dried at 105 °C for 24 h and stored in a desiccator for subsequent uses [31]. 

### 2.3. Characterization of Activated Porous Carbon

The porous morphology of the AFAC was observed using SEM (S-4800F FE-SEM, HITACHI, Japan. The surface area, pore-volume, and pore size distribution were also measured with Tristar-3000 surface area and porosity analyzer (Micromeritics, Norcross, GA, USA) at −196 °C. Before measuring the surface area, the sample was degassed at 200 °C for 12 h. 

### 2.4. Point of Zero Charges (pH_pzc_) Measurement

The AFAC pH_pzc_ characteristics were measured following the solid addition method. In this method, 25 mL of 0.2 M KNO_3_ was placed in different flasks, and their pH was adjusted between 2 and 12 by adding 0.2 M HCl. The volume of the solution was adjusted to 50 mL by adding the KNO_3_ solution. Then, 0.5 g of AFAC was added to each flask, and the suspension was placed on a shaker for 24 h. After equilibrating for 1 h, the supernatant’s final pH (pH_f_) was noted. 

### 2.5. Batch Adsorption Study 

A 1000 mg/L AZM stock solution was prepared and used to prepare the other concentrations, i.e., 25–200 mg/L, by dilution. Each batch adsorption experiment was carried in a 150-mL flask containing 100 mL of AZM solution and placed in a thermostatic shaker (CF0201003, Guangdong, China) (120 rpm) bath. The pH was controlled by adding 0.1 N NaOH or HCl. The adsorbent dosage was varied from 0.1 to 1.5 g/L, corresponding to an adsorbent to adsorbate ratio of 0.5 to 60. The temperature was varied from 273 to 325 K to evaluate the adsorption thermodynamics. Moreover, the adsorption kinetics were analyzed by measuring the AZM concertation at different time intervals up to 120 min. The concentration of AZM was measured using high-performance liquid chromatography (HPLC) (Agilent Technologies, Santa Clara, CA, USA) with Eclipse XDB-C18, 5 μm (4.6 × 150 mm) column, and UV detector operated at 230 nm with 1 mL/min of 40/60 acetonitrile/water as the mobile phase.

The AZM removal percentage (*R*) and adsorption capacity (*q*_e_, mg/g) are calculated as follows [17]:(1)$$R=\frac{\left(C_{0}-C_{t}\right)}{C_{0}}\times 100$$
(2)$$q_{e}=\frac{\left(C_{0}-C_{t}\right)\times V}{m}$$
where *C*_0_ is the initial AZM concentration (mg/L), *C*_t_ is AZM concentration (mg/L) at time t, *V* is the solution volume (L), and *m* is the AFAC (g). The Langmuir adsorption model, Temkin model and other associated calculations are also provided in the Appendix A and Appendix A. 

## 3. Results and Discussions

### 3.1. Characteristic of AZM and Characterization of Adsorbent AFAC 

AZM is poorly soluble in water, but these antibiotics are widely used against various infections where different pathogens infect them. Azalide-type drugs, such as AZM, are recognized as pharmaceutical pollutants when the drug is drained into the water after use. Meanwhile, few approaches were employed to remove this drug from the aqueous wastewater, but are found to be very inefficient and ineffective [16,17,18], as the removal efficiency largely depends on the important factors, including the physicochemical characteristics of AZM, type, and nature of the solvent, and approached used for this purpose. The drug AZM anhydrous has the following physiochemical characteristics (aqueous solubility ~ 0.514 mg/mL; logP ~ 3.0 [16,17,18]).

The purpose of preparing a high surface area porous adsorbent is to ensure fast and efficient adsorption of the antibiotics into the porous network. Figure 1a shows the SEM image of the AFAC before adsorption, revealing the type of porous structure of AC derived from AF. As shown in Figure 1a, the particles from AFAC are in irregular pore sizes and shapes but are porous, revealing that AFAC could be used as the potential adsorbent. The structure of AFAC post-adsorption (Figure 1b) is different from the structure shown in Figure 1a, due to the loading of antibiotics into the porous network of AFAC. The hysteresis loops confirm that the N_2_ adsorption-desorption isotherm of this porous structure is consistent with type IV isotherm, corresponding to mesoporous solid (Figure 1c). The inset in Figure 1c shows the pore size distribution, with an average pore size of 4.9 nm. The isotherm suggested that the adsorbent contains micropores and mesopores as the relative pressure (P/P_0_) at 0.4 to 0.65, which could be ascribed to the sample’s mesoporous nature [31]. In contrast, low-relative pressure (P/P_0_) at 0.4 could be responsible for micropores.

The AFAC nitrogen adsorption-desorption results are also outlined in Table 1, indicating that the AFAC has a specific surface area of 484.1 m^2^/g, pore volume of 0.47 cm^3^/g, a porosity of 53.4%, bulk density of 0.245 g/cm^3,^ and average pore size of ~ 5 nm. The results of EDX are shown in Table 1. As you can see, the elements carbon, oxygen, magnesium, potassium, and hydrogen are almost the main elements of AFAC, and the negligible amounts of sodium and iron were observed in the final composition of AFACHere; the yield (Y) of the prepared AFAC was calculated as follows [32]:(3)$$Y=\frac{W_{\mathrm{AC}}}{W_{\mathrm{AF}}}\times 100$$

### 3.2. Adsorption of AZM on AFAC

#### 3.2.1. Effect of Temperature and Contact Time 

The adsorption time and temperature effects were investigated under the optimum pH of 9, AFAC dose of 1 g/L, and 100 mg initial AZM concentration. The adsorption capacity and removal efficiency of AZM by the AFAC adsorbent is shown in Figure 2. The adsorption process is very rapid at the initial stage (0–30 min) due to the adsorbent’s high surface area and large pore volume. This initial stage is followed by a slower stage (30–60 min), and finally, an equilibrium stage is reached after ~75 min. The rapid removal rate at the first stage is due to the high concentration of the active adsorption sites on the AFAC surface in the early adsorption process [33]. These active sites become fully occupied by the adsorbed AZM molecules as the length of contact time between the adsorbent and the adsorbate increases, and the repulsive force that occurs between the AZM molecules on the surface of adsorbents and AZM molecules in the bulk liquid phase reduces the AZM adsorption rate [34,35].

The high AZM adsorption rate is due to the higher probability of interactions between AZM ions and AFAC porous particles [32] and the presence of more active pores. These pores can accommodate more AZM on their surface. The rapid uptake of AZM ions by AFAC is one of the parameters which could be potentially considered for economic wastewater treatment plant applications [33].

#### 3.2.2. Effect of Adsorbent Dose 

The impact of AFAC dosage on the AZM adsorption capacity and removal efficiency was investigated using aqueous by changing the AFAC adsorbent dosage between 0.1 to 1.4 g/L. All the other adsorption parameters, i.e., pH, temperature, and contact time, were constant. The results in Figure 3 show that the AZM adsorption capacity decreased from 297.4 to 63.8 mg/g, with the adsorption dosage of 0.1 and 1.4 g/L, respectively. This decrease in adsorption capacity with the AFAC dosage is attributed to the decrease in the adsorbate/adsorbent ratio with the increase in dosage consistent with previously reported results [34,35]. On the other hand, the % removal increases with the AFAC dosage and reaches a near plateau at a 1 mg/mL dosage. Therefore, 1 mg/mL AFAC was selected as the optimum dosage used in the following experiments. 

Consequently, it was found that the availability of a larger surface area and more adsorption sites already accelerated with an increase in the adsorbent dose [36,37]. Increasing the adsorbent dose over 1 mg/mL resulted in a minor change in the removal percentage. The establishment of a dense screening layer at the adsorbent surface because of the accumulation of AFAC particles and a decrease in the distance between the AFAC molecules, known as the screening effect, happens with a higher adsorbent dose, could explain this phenomenon [38]. The binding sites were hidden from AZM molecules by the condensed layer on the adsorbent surface. Furthermore, because AFAC overlapped, AZM molecules competed for a limited number of accessible binding sites [39]. Agglomeration or aggregation at higher AFAC dosages lengthens the diffusion channel for AZM adsorption, lowering the adsorption rate [40].

#### 3.2.3. Effect of pH 

The effect of solution pH on the adsorption capacity of AFAC is explained by analyzing the AZM surface charges and dissociation constant (pKa) [35]. The oxygen functional groups on the adsorbent surface can also impact the adsorbent performance related to the pH_(pzc)_. In other words, the H^+^ or OH^−^ ions in the solution change the adsorbent’s surface charge and ultimately impact the antibiotics removal efficacy [36]. 

Figure 4a shows the effect of pH on AZM adsorption by AFAC. The pH_pzc_ of AFAC is shown in Figure 4b. At pH lower than pH_pzc_, the surface functional groups deprotonate by the presence of OH^−^ ions in the solution [37]. At pH higher than the pH_pzc_, the surface functional groups deprotonate by the H^+^ ions in the solution [38]. The surface of AFAC is positively charged at pH lower than pH_pzc_, i.e., 8.45.

On the other hand, the pKa of AZM is ~8.6–9.5; hence, in an acidic medium, AZM is protonated. As a result, the adsorption capacity of AFAC in pH < 3 is low due to electrostatic repulsion between AZM and AFAC, and competition between H^+^ with AZM for adsorption on AFAC active sites. Accordingly, the electrostatic attraction between AZM and the AFAC surface gradually increases to yield the highest adsorption at pH = 9–11.

#### 3.2.4. Adsorption Isotherms 

The adsorption isotherm for removing AZM ions from the aqueous solution was obtained at equilibrium conditions to establish the relationship between the adsorbed AZM amount onto the AFAC, and that amount remained in the aqueous solution [39]. The experimental data were examined using four different isotherm models, i.e., Langmuir, Freundlich, Temkin, and Dubinin–Radushkevich (D-R) [40,41,42]. 

The Freundlich model is an empirical isotherm that describes multilayer adsorption on a heterogeneous system. It is mathematically expressed by Equation (4) [43]:(4)$$\begin{array}{cc} ln\;q_{e}=ln\;K_{F}+\frac{1}{n}ln\;c_{e} & n>1 \end{array}$$
where *n* and *K_F_* (mg/g) are the Freundlich Isotherm constants related to adsorption intensity and capacity. *n* and *K_F_* are calculated from the slope and intercept of the linear fit to the $lnq_{e}$ vs. $lnc_{e}$_._ A slope ranging between zero and unity indicates the heterogeneity of the surface and the adsorption intensity, as the slope is very low (near zero). A slope near unity refers to a chemisorption process, where 1/*n* higher than unity implies cooperative adsorption. The Freundlich isotherm constant relates to the favorability of the adsorption process. The adsorption process is favorable when 1 < *n* < 10 [44]. 

The results in Figure 5 show that Freundlich isotherm reasonably describes the adsorption of antibiotics to AFAC. The value of n for AZM adsorption (Table 2) is in the range of 1 < *n* < 10, indicating favorable adsorption. The Freundlich constant, *K_F_*, signifies the strength of the interactions between the adsorbate and the adsorbent. The obtained *K_F_* ranged from 3.08 at 273 K to 58.95 at 333 K, indicating enhanced interactions at higher temperatures. It is worth noting that the three other isotherms did not provide any reasonable fit to the adsorption results, as shown by the very low R^2^ provided in Table 2. 

The maximum measured q_m_ was 374 mg/g, obtained at 333 K and a dose of 0.1 g/L. This maximum adsorption capacity was much higher than the reported capacity for antibiotics, using a variety of adsorbents, as shown in Table 3. This indicates the potential of the AFAC as an adsorbent for large-scale applications.

#### 3.2.5. Adsorption Kinetic 

The kinetic of the AZM adsorption on AFAC is also assessed using the pseudo-first-order kinetic model, the pseudo-second-order model, and the intra-diffusion model. The integrated form of the pseudo-first-order kinetic equation is given by [45]:(5)$$ln\left(q_{e}\;–\;q_{t}\right)=ln\;q_{e}\;–\;k_{1}t$$
where *q_e_* is the equilibrium sorption uptake, *q_t_* (mg/g) is the amount of adsorbed AZM on the AFAC at time *t*, and *k*_1_ (1/min) is the rate constant of the first-order adsorption. *q_e_* is extrapolated from the experimental data at time *t* = 0. A straight line fitting of Ln (*q_e_* − *q_t_*) versus *t* confirms the pseudo-first-order kinetic.

The analysis of the adsorption date (Figure 6) revealed that the adsorption process follows the pseudo-first-order kinetics, with a rate constant that ranges from 0.046 to 0.055 min^−1^, indicating a weak temperature dependence. The analysis of the results using second-order and intra-particle diffusion models showed a poor correlation between the results and these models, as summarized in Table 4. Other details of the analysis are included in the Appendix A.

In the intraparticle diffusion (IPD) kinetic model is described by Equation (6) [46]:(6)$$q_{t}=k_{P}t^{1/2}+C$$
where *k_p_* is the IPD rate constant $\left(\frac{\mathrm{mg}}{g\cdot \min^{0.5}}\right)$ and *C* is a constant that describes the initial adsorption capacity. When IPD governs the adsorption process, a linear plot is obtained by drawing *q_t_* versus *t*^1/2^ (in Figure 7), and IDP is addressed as a rate-controlling step when the obtained lines pass through the origin. In our case, the kinetic data cannot be described by a single IPD model (Figure 7a), but by the three-stage process, as shown in Figure 7b. As shown in Table 5, the first stage (t < 30 min) is very rapid, indicated by the large rate constant, $k_{d_{1}}$. This adsorption mechanism during this stage is associated with adsorption on the external surface. The second stage (t = 30–60 min) has a moderate rate constant ($k_{d_{2}}$). During this second stage, the adsorption process proceeds via adsorption on the internal mesopores. The last stage (t > 60 min) is a very slow process, as indicated by the very low rate constants ($k_{d_{3}}$). The adsorption could be attributed to the chemical adsorption step on the remaining active sites.

#### 3.2.6. Adsorption Thermodynamics

To study the thermodynamic of the adsorption process, batch adsorption experiments were carried at 273, 288, 303, 318, and 333 under the optimum adsorption conditions. The AZM adsorption capacity has increased with temperature. For example, the capacity has increased from 76.13 mg/g at 273 K, to 99.44 mg/g at 333 K. This is attributed to the high kinetic energy of AZM cations at higher temperatures; causing sufficient contact between the AZM and the AFAC active sites, and leading to an increase in the removal efficiency [47]. These results also indicate that the adsorption is a physical rather than a chemical process. A similar trend has been observed in other studies to remove pollutants from the aqueous phase [48,49]. Moreover, increasing the temperature may increase the pore size significantly, affecting the porous carbon uptake capacity [50].

Thermodynamic parameters such as the change in free energy (Δ*G*°), enthalpy (∆*H*°), and entropy (∆*S*°) were calculated using Equations (7) and (8) [51,52]:(7)$$\Delta G_{0}=-RT\ln K$$
(8)$$\ln K_{L}=\frac{\Delta S^{{}^{\circ}}}{R}-\frac{\Delta H^{{}^{\circ}}}{RT}$$
where *T* (K) is the absolute temperature, *R* (8.314 KJ/mol·K) is the ideal gas constant, *K_L_* (L/mol) represents the Langmuir constant. ∆*H*° and ∆*S*° are calculated from the slope and intercept of the Vant Hoff plot of Ln *K_L_* versus (1/T). The thermodynamic parameters of the adsorption process are provided in Table 6. ∆*G*° was negative over the entire temperature range, revealing the feasible and spontaneous adsorption process in the 273–333 K studied temperature range. As physical adsorption process has −20 kJ/mol < Δ*G*° < 0 kJ/mol, while chemical adsorption has −400 kJ/mol < Δ*G*° < −80 kJ/mol [53]. In this study, ∆*G*° was between −2.38 and −8.72 kJ/mol, confirming the physical nature of the adsorption process. 

The positive Δ*H*° indicates that the adsorption process is endothermic. The diffusion coefficient of AZM molecules increases with temperature, enhancing the adsorption. The positive Δ*S*° confirms that the process is spontaneous, with a positive binding affinity of AZM on the AFAC active sites. 

## 4. Conclusions 

We have successfully prepared the AFAC with a specific surface area of 484.1 m^2^/g, pore volume of 0.472 cm^3^/g, and porosity of 53.4% using Azolla filiculoides (AF) as an activated porous carbon source for the removal of antibiotic AZM as a pollutant. The porosity of AFAC was confirmed by SEM and BET isotherms. Then, isotherms, kinetic and thermodynamic studies were conducted for the AZM antibiotic removal using AFAC. The characteristics of the AFAC and its performance for the adsorption of AZM from an aqueous solution were investigated, and the results have suggested that the percentage removal was dependent on temperature, pH, adsorbent dosage, and contact time. AFAC has exhibited a high adsorption capacity for antibiotic AZM, and the antibiotic was effectively removed from the aqueous solution used. For example, maximum removal was 97.9% at optimum pH of 9, contact time 75 min, and adsorbent dose 1 g/L. The negative Δ*G*° and positive Δ*H*° and Δ*S*° confirmed that the adsorption of AZM on AFAC was spontaneous and endothermic. The Freundlich isotherm has described the equilibrium adsorption results well. Our results suggested that AFAC is an efficient and low-cost adsorbent for removing antibiotics from wastewater.

## Data Availability

The data presented in this study are available on request from the corresponding authors.

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
