# Peer review of "Adsorptive Removal of Azithromycin Antibiotic from Aqueous Solution by Azolla Filiculoides-Based Activated Porous Carbon"

_nanomaterials, 2021, doi:10.3390/nano11123281_

Round 1

Reviewer 1 Report

Please find the attached file for the comments

Author Response

Reviewer 1:

The submitted article entitled with “Adsorptive Removal of Azithromycin Antibiotic from Aqueous Solution by Azolla Filiculoides based Activated Porous Carbon” addressed an exhaustive study for remove AZM from contaminated water using biological method. The work is well designed and well written with informative data. I consider for publication after minor revision

Response: We appreciate your constructive comments.  

Comment 1: In introduction section, authors should include physicochemical properties of AZM such as its aqueous solubility, logP value, and pKa value with suitable references such as (DOI: 10.1007/s11356-021-15031-w and doi10.1016/j.jwpe.2021.101973. Authors should update recent reports published to treat contaminated water with azithromycin using green nanoemulsion technique in introduction section. Please see highlighted lines in yellow.

Response:  We have added the suggested references (Ref 17 and 18) and modified the discussion in the introduction section (66 and 67) and in section 3.1 (lines 141-150). These changes are highlighted in yellow.

Comment 2: In this article authors used AF alga to treat AZM contaminated water. Did authors study the specificity of the developed method for AZM or related substance present in trace amount? Is the reported method feasible to set up for large scale treatment in healthcare system?

Response. In this manuscript, the focus was on the synthesis of AFAC and investigating its ability to remove AZM. We agree with the author about the importance of the selectivity investigation.  However, the selectivity of the adsorbent to AZM versus other pollutants is beyond the scope of this manuscript. In a subsequent phase, the adsorbent selectivity and breakthrough adsorption experiments will be carried to assess its feasibility for treatment of real wastewater steams and large-scale operation. Nonetheless, the preliminary results in this manuscript (the capacity and the rapid adsorption) and the low cost of the material suggest potential of the adsorbent for large-scale operation.

Comment 3: In equation 1 and 2, the symbol must be italic and subscript need to be nonitalic to keep uniformity. Authors are suggested to remove the word “Dosage”. It must be “dose”.

Response:  As suggested, we have revised equations 1 and 2 and related terms as well. We have removed word “dosage” from lines 138 and 139.

Comment 4: Figure 1a and b must be replaced with image revealing caption (voltage, magnification and image information). In figure 1c, the pore size should be written as 5.0 nm (avoid to use 4.9 nm as size diameter). In Table 1, please match the value of pore volume mentioned in text and table.

Response: We thank the reviewer for the comments. Figures 1a and b are replaced with new figures and the scale-bar is clearly shown on the image. The average size is rounded to 5 nm. The pore volume in Table 1 (now inset of Figure 1c) and the text are corrected too.  Please see Figure 1.  

Comment 5: In page 6, line 173, authors mentioned “The high AZM adsorption rate is due to the higher probability of interactions between AZM ions and AFAC porous particles”. Which types of interaction?. Figure 5 is not clear. Use image of high resolution with clear label and scale on axis. Please mention the unit in “273, 288, 303, 318, and 333 K”

Response: Sorry that you have missed our interpretation. Please see highlighted sections and lines in yellow (lines 187-193, 213-223, 309-322). Figure 5 is also revised. We updated Figure 5.

Reviewer 2 Report

Reviewer’s comments

 Manuscript Number: nanomaterials-1415114

Title: Adsorptive Removal of Azithromycin Antibiotic from Aqueous Solution by Azolla Filiculoides based Activated Porous Carbon

Journal: Nanomaterials

The authors present a manuscript focusing on the preparation and characterization of activated carbon and its application for Azithromycin Antibiotic removal. Here are some comments that need to be addressed:

  1. Since the AC preparation involved ZnCl2 addition, EDX analysis is needed to show the elemental composition of the AC to make sure that all Zn content is removed by washing.
  2. High magnification SEM images should be provided to show the porosity and more morphological details. Where there is no porosity is observed in the current images although it is mentioned in the title “Activated Porous Carbon”
  3. The adsorption mechanism (how azithromycin antibiotic binds with azolla filiculoides based on activated porous carbon) should be discussed.
  4. Equations 1 and 2 need to be supported with relevant references. The following references may help Physica E 2019; 106: 150-155, J. Mol. Liq. 2017; 237: 466–472, J. Mol. Liq. 2016; 218: 191-197.
  5. The discussion is very simple and surface. A deep discussion is needed as well as the adsorption findings should be correlated to the structural and morphological results.
  6. The removal efficiency of AC from Azolla Filiculoides should be compared with other waste-derived AC adsorbents to show the advantages of using this material. The following literature may help: Desalination and Water Treatment, 84 (2017) 205-214, Journal of Wuhan University of Technology-Mater. Sci. Ed., 32 (2017) 305-320, Jurnal Teknologi, 79(7) (2017) 1-10, Malaysian Journal of Analytical Sciences, 21 (2) (2017) 334-345.

Based on the comments given above, I can say that it cannot be considered for publication in Nanomaterials in its present form. Therefore, Major Revision is needed before resubmission.

Author Response

Reviewer 2:

Manuscript Number: nanomaterials-1415114

Title: Adsorptive Removal of Azithromycin Antibiotic from Aqueous Solution by Azolla Filiculoides based Activated Porous Carbon

Journal: Nanomaterials

The authors present a manuscript focusing on the preparation and characterization of activated carbon and its application for Azithromycin Antibiotic removal. Here are some comments that need to be addressed:

  1. Since the AC preparation involved ZnCl2 addition, EDX analysis is needed to show the elemental composition of the AC to make sure that all Zn content is removed by washing.

Response: EDX analysis was carried, and the results are presented in Table 1. The main elements are C (54%), O (38%), and H2 (3.7%),  in addition to small concentration of Mg (3.4%), and H (1.7%), and traces of Fe. No peaks for Zn or Cl are observed.

  1. High magnification SEM images should be provided to show the porosity and more morphological details. Where there is no porosity is observed in the current images although it is mentioned in the title “Activated Porous Carbon”

Response: A new and high-quality images of Figures 1a and b are provided. Please see revised Figures 1a and 1b.

  1. The adsorption mechanism (how azithromycin antibiotic binds with azolla filiculoides based on activated porous carbon) should be discussed.

Response: The absorption mechanism is discussed in the revised considering the additional “intra particle diffusion” mechanism. Please see highlighted sections and lines in yellow (lines 187-193, 213-223, 309-322).

  1. Equations 1 and 2 need to be supported with relevant references. The following references may help Physica E 2019; 106: 150-155, J. Mol. Liq. 2017; 237: 466–472, J. Mol. Liq. 2016; 218: 191-197.

Response: References for equations 1 and 2 are cited in the revised manuscript.

  1. The discussion is very simple and surface. A deep discussion is needed as well as the adsorption findings should be correlated to the structural and morphological results.

Response: We have revised the discussion section in response to the reviewers’ comments. Please see highlighted sections throughout the revised manuscript, for examples lines 187-193, 213-223, 309-322).

  1. The removal efficiency of AC from Azolla Filiculoides should be compared with other waste-derived AC adsorbents to show the advantages of using this material. The following literature may help: Desalination and Water Treatment, 84 (2017) 205-214, Journal of Wuhan University of Technology-Mater. Sci. Ed., 32 (2017) 305-320, Jurnal Teknologi, 79(7) (2017) 1-10, Malaysian Journal of Analytical Sciences, 21 (2) (2017) 334-345.

Response: The suggested references are cited, and A new Table (Table 3) is also added to compare the performance of the AFAC to other adsorbents.

Based on the comments given above, I can say that it cannot be considered for publication in Nanomaterials in its present form. Therefore, Major Revision is needed before resubmission.

Response: We believe the revised version has improved and all comments are addressed to make the manuscript acceptable for publications.

Reviewer 3 Report

Dear Authors, I appreciate the work you put into the article. The paper seems to be well organized and readable. Nevertheless, there is still lots of work to do.

This is not surprising that carbons are able to adsorb. The material you received is not shocking, rather in the middle; exact characterization (not only SEM!) is obligatory, this is not standard material. C.a. 2% moisture could be extremely important!

Isn’t it obvious that when the adsorbent increases the qe decreases? More, this experiment needs to be repeated, you cannot exceed 40%! Otherwise, additional effects are decisive.

Additionally, there is one question & one request.

Question: where is “nano”? (neither adsorbate nor adsorbent they are not), I would suggest shifting to a more adequate journal e.g. C­–carbon also MDPI

Request (and warning): please avoid further papers in the style like: carbon from... another plant for adsorption of... another drug, this is not good science.

Carbons are perfect adsorbents via their lack of clear selectivity during the process

I agree that removing antibiotics from wastewater is a challenge, but your results do not suggest that AFAC is an efficient and low-cost adsorbent, sorry. The plant is cheap but other reagents and carbonization are rather not. What with Zn and HCl after carbonization, is it an ecologic process??  

Author Response

Reviewer 3:

Dear Authors, I appreciate the work you put into the article. The paper seems to be well organized and readable. Nevertheless, there is still lots of work to do.

Response: We thank the reviewer and appreciate the constructive comments. 

This is not surprising that carbons are able to adsorb. The material you received is not shocking, rather in the middle; exact characterization (not only SEM!) is obligatory, this is not standard material. C.a. 2% moisture could be extremely important!

Response: Unfortunately, there was a typo-mistake in the value of the water content. The correct water content is 0.2%, see updated Table 1.

Isn’t it obvious that when the adsorbent increases the qe decreases? More, this experiment needs to be repeated, you cannot exceed 40%! Otherwise, additional effects are decisive.

Response: The removal efficiency is related to the adsorption capacity, qe,

. Therefore, as pointed by the reviewer, the removal capacity at the lowest dose (0.1 mg/ml) must be between 22 and 40% depending on temperature. However, the % removal increases with dose. For example, at a dose of 1 mg/ml, i.e., 1 g/L, the %removal is higher dose, . These are consistent with the results shown in Figures 2 and 3.

Additionally, there is one question & one request.

Question: where is “nano”? (Neither adsorbate nor adsorbent they are not), I would suggest shifting to a more adequate journal e.g., C­–carbon also MDPI

Response: The activated carbon adsorbent in this study is a mesoporous material (average pore size is 5 nm).  This is consistent with the scope of Nanomaterials (“with characteristic mesoscopic properties”). Therefore, many activated carbon-based studies have been published in Nanomaterials. For example, here are some very recent (2020 and 20201) publications in Nanomaterials:

  1. Benítez et.al., "Pistachio Shell-Derived Carbon Activated with Phosphoric Acid: A More Efficient Procedure to Improve the Performance of Li–S Batteries," Nanomaterials, vol. 10, no. 5, p. 840, 2020.
  2. Jakubec et.al., "Flax-Derived Carbon: A Highly Durable Electrode Material for Electrochemical Double-Layer Supercapacitors," Nanomaterials, vol. 11, no. 9, p. 2229, 2021.
  3. Kim et.al. "Bamboo-Based Mesoporous Activated Carbon for High-Power-Density Electric Double-Layer Capacitors," Nanomaterials, vol. 11, no. 10, p. 2750, 2021.
  4. Lee et.al. "Effect of Mesopore Development on Butane Working Capacity of Biomass-Derived Activated Carbon for Automobile Canister," Nanomaterials, vol. 11, no. 3, p. 673, 2021.
  5. Liao et.al. "Modified Camellia oleifera Shell Carbon with Enhanced Performance for the Adsorption of Cooking Fumes," Nanomaterials, vol. 11, no. 5, p. 1349, 2021.
  6. Men’shchikov et.al. "Thermodynamics of Adsorbed Methane Storage Systems Based on Peat-Derived Activated Carbons," Nanomaterials, vol. 10, no. 7, p. 1379, 2020.
  7. Men’shchikov et al., "Thermodynamic Behaviors of Adsorbed Methane Storage Systems Based on Nanoporous Carbon Adsorbents Prepared from Coconut Shells," Nanomaterials, vol. 10, no. 11, p. 2243, 2020.
  8. Shrestha et.al. "Efficiency of Wood-Dust of Dalbergia sisoo as Low-Cost Adsorbent for Rhodamine-B Dye Removal," Nanomaterials, vol. 11, no. 9, p. 2217, 2021.
  9. Shulga et al., "The Concentration of C(sp3) Atoms and Properties of an Activated Carbon with over 3000 m2/g BET Surface Area," Nanomaterials, vol. 11, no. 5, p. 1324, 2021.

Request (and warning): please avoid further papers in the style like: carbon from... another plant for adsorption of... another drug, this is not good science.

Response: The antibiotics are emergent pollutants and investigating their removal from water is of significant interest. Moreover, preparation of adsorbent from available and cheap sources is desired.

Carbons are perfect adsorbents via their lack of clear selectivity during the process

Response: Activated carbon are one of the perfect adsorbents for the removal of pollutants, and the innovation is created by using inexpensive adsorbents which can increase this efficiency and reduce costs potentially.

I agree that removing antibiotics from wastewater is a challenge, but your results do not suggest that AFAC is an efficient and low-cost adsorbent, sorry. The plant is cheap but other reagents and carbonization are rather not. What with Zn and HCl after carbonization, is it an ecologic process??  

Response: One of the main environmental problems in wetlands is the collection of aquatic plants, which has created many dangers for aquatic animals. Therefore, collecting it and removing the pollutant will solve these two environmental problems. Every year, a significant budget is spent on the elimination of Azolla in different countries to preserve aquatic organisms, and therefore, this cost will also be saved. EDX shows absence of Zn whereas sample is washed with the copious amount of water that will remove HCl. 

Round 2

Reviewer 1 Report

The revision is satisfactory for publication.

Author Response

Reviewer 1 Comments:

The revision is satisfactory for publication.

Response: We thank the reviewer for the positive feedback.

Reviewer 2 Report

Reviewer’s comments

Manuscript Number: nanomaterials-1415114

Title: Adsorptive Removal of Azithromycin Antibiotic from Aqueous Solution by Azolla Filiculoides based Activated Porous Carbon

Journal: Nanomaterials

The authors have addressed most of the comments in the revised version of the manuscript. Thus it could be accepted for publication.

Author Response

Reviewer 2 Comment:

The authors have addressed most of the comments in the revised version of the Manuscript. Thus it could be accepted for publication.

Response: We thank the reviewer for the positive feedback.

Author Response

At this point, we do not wish to continue arguing with Reviewer 3. We attached the reviewer's comments on the original Manuscript, our response to these comments, and the reviewer's response to our response. We kindly request the Editor review our responses to his comments on the original Manuscript and his un-based rejection of our response and decide accordingly.
